# Quantitative Aspects of the Human Cell Proteome

**DOI:** 10.3390/ijms24108524

**Published:** 2023-05-10

**Authors:** Stanislav Naryzhny

**Affiliations:** 1Institute of Biomedical Chemistry, Pogodinskaya Str. 10, 119121 Moscow, Russia; snaryzhny@mail.ru; 2Petersburg Institute of Nuclear Physics (PNPI) of National Research Center “Kurchatov Institute”, 188300 Gatchina, Russia

**Keywords:** human proteome, quantitation, formula

## Abstract

The number and identity of proteins and proteoforms presented in a single human cell (a cellular proteome) are fundamental biological questions. The answers can be found with sophisticated and sensitive proteomics methods, including advanced mass spectrometry (MS) coupled with separation by gel electrophoresis and chromatography. So far, bioinformatics and experimental approaches have been applied to quantitate the complexity of the human proteome. This review analyzed the quantitative information obtained from several large-scale panoramic experiments in which high-resolution mass spectrometry-based proteomics in combination with liquid chromatography or two-dimensional gel electrophoresis (2DE) were used to evaluate the cellular proteome. It is important that even though all these experiments were performed in different labs using different equipment and calculation algorithms, the main conclusion about the distribution of proteome components (proteins or proteoforms) was basically the same for all human tissues or cells. It follows Zipf’s law and has a formula N = A/x, where N is the number of proteoforms, A is a coefficient, and x is the limit of proteoform detection in terms of abundance.

## 1. Introduction

To better understand the functionality of the human proteome, we need information about the abundance of all proteome components—protein complexes, proteins, and proteoforms. Identification and quantification of the proteome’s components in different human tissues provide a valuable resource for understanding the multiple processes performed [1]. The situation here is not straightforward because of the complexity of proteins themselves. This complexity may arise from allelic variations, alternative splicing of RNA transcripts, and post-translational modifications (PTMs). All these cellular events create distinct proteoforms that modulate a wide variety of biological processes [2,3]. The term “proteoform” encompasses all sources of biological variation that alter a protein’s primary sequence and composition [2]. 

So far, it has been impossible to identify and calculate all proteoforms presented in a single human cell or in the human plasma [1,3]. The main problem is the vast dynamic range of abundance, in which the number of copies of different proteoforms is from one to a billion molecules [4]. A combination of experimental and bioinformatics approaches has been utilized to try to solve this problem [5,6]. Considering the heterogeneity of the human population, it seems that detailed knowledge about all human proteoforms is a task for the future. But based on genomics and proteomics data obtained from analysis of different cells or tissues, it is possible to make some conclusions. The starting point here is the number of genes. Currently, the data support 19,750 protein-coding genes in humans (https://www.hupo.org/, accessed on 10 December 2022). However, not all of these are expressed in a particular tissue or a cell line. Around 8,000 genes have been observed to be expressed in all tissues and cell lines, and the number of the genes expressed in most human tissues can be from 11,000 to 13,000, except in the testis, where this number is ~15,000 [7]. However, these numbers should be doubled when considering splice variants [8]. Important sources of genetic variation include single-nucleotide polymorphisms (SNPs) and mutations. Finally, many human proteins undergo PTMs, such as glycosylation, phosphorylation, acetylation, among a few hundred others, which give rise to more proteoforms [2]. Though many proteins are unmodified, some, like histones, are already annotated with hundreds of modifications [9]. Fortunately, extreme complexity can be avoided due to the high degree of control over the enzymatic writing and maintenance of PTMs—not all theoretical proteoforms are actualized [2,10].

This review presents an analysis of available experimental information about the amount and abundance distribution of proteins/protein groups/proteoforms in human cells and tissues.

## 2. Quantification

### 2.1. Panoramic Quantification

Panoramic quantitation can be done based principally on two approaches—mass-spectra analysis or densitometry of 2DE images. For MS quantitation, we assume that the measured signal has a linear dependence on the amount of material in the sample. MS-based quantitation itself can be performed using two different strategies: untargeted global quantification of thousands of proteins and targeted quantification of only a few components [11,12,13,14]. The untargeted quantification can be further divided into two subgroups: label-based quantification utilizing stable isotopes and label-free quantification [15]. This quantification can be relative or absolute. Using relative quantification, it is possible to compare the amount of single proteins or whole proteomes in different samples. Conversely, absolute quantification provides information regarding the total amount or concentration of proteins within a model. 

As the measurements are based on signals from the peptides that are generated by the specific hydrolysis of polypeptides, protein inference is a significant problem in bottom-up protein quantification. High-throughput approaches for proteoform analysis are based on “top-down” mass spectrometry, in which the whole polypeptide (proteoform) is measured [16,17,18,19,20]. However, this method has mass limitations due to the capabilities of the instruments, and quantification is technically challenging—only relative quantification of proteins can be performed [20]. As the average mass of a human protein is ∼60 kDa, alternate methods are necessary to identify and quantify all proteoforms at high throughput [21]. 

Currently it is only possible to perform the panoramic quantitative analysis of all proteoforms by pre-separation using 2DE [22]. Coomassie blue protein staining can be used for reliable quantitative abundance estimation. The linear properties of this dye in a wide range of concentrations are utilized in a Bradford assay for protein measurement [23]. In gels, the linear range of Coomassie is from ~10 ng to 20 µg [24,25]. Using a scanner with linear response over the 0–3.0 absorbance range, it is possible to quantitate all spots in the 2DE gel [24] (Figure 1). The weak points are sensitivity and the chance of the presence of different proteoforms in the same spot [26,27]. However, the combination of 2DE with bottom-up MS (ESI LC-MS/MS) can solve this problem. An additional level of panoramic analysis of proteoforms can be reached using a sectional 2DE approach (sec2DE) (Figure 1) [26,27]. We use this approach to kill two birds with one stone—identification and quantification of proteoforms. 

We should remember an important point regarding the quantification of proteins. In all cases of protein quantification, the term “abundance” is used, but it can have different meanings. In the case of spot staining, abundance is measured by the concentration of protein mass in a spot. However, when using MS, the abundance is proportional to the molar concentration (number of protein molecules, copy number). For instance, two proteins, vimentin (MW 54,000) and cofilin (MW 18,000), have the same copy numbers estimated by MS (emPAI, iBAQ, etc.). However, in the case of 2DE, the spot intensity of vimentin will be 3 times higher than the spot intensity of cofilin, as each molecule of vimentin is three times the size of the cofilin molecule. Accordingly, the level of vimentin measured by staining will be three times higher than that of cofilin. This must be kept in mind when looking for a level of a particular protein.

### 2.2. Aspects of Cell Size

A human body contains, on average, 3.7 ± 0.8 × 10^13^ cells (BNID 109716, (https://bionumbers.hms.harvard.edu/search.aspx, accessed on 28 February 2023) [29], plus a similar number of resident microorganisms (human microbiome) [30,31]. Among these are over 200 different types of human cells that execute various functions. Thus, these cells are called ‘specialized’. Six of the cell types comprise 97% of human cells: red blood cells (71%), glial cells (8%), endothelial cells (7%), dermal fibroblasts (5%), platelets (4%), and bone marrow cells (2%). Other cells account for the remaining 3% [29,31]. It is interesting that ~90% of the human cells are enucleated. They are mostly red blood cells and platelets circulating in the blood vessels. Conversely, in terms of mass, muscle and fat account for 75% of body weight, although muscle and fat cells are quite large and make up only 0.1% of the total cell number. The sizes of cells span a large range, as shown in Table 1 [30]. The typical (average) size of the human nucleated cell is 20–40 µm, with a volume of 2000–4000 μm^3^. For instance, hepatocytes are polyhedral cells with a volume of ~3.4 × 10^3^ μm^3^ [31,32]. 

The amount of protein mass in a cell depends on the cell size. The protein content scales roughly linearly with cell volume, and the copy numbers of the majority of proteins are proportional to the cell volume [34,35]. There are approximately the same number of proteins per cell volume in different human cells [35]. These numbers are similar in bacteria (*E. coli*), yeast, and human cell lines (HeLa) [35].

This situation was used for generation of the so-called “proteomic ruler”, in which histone copy numbers are used for normalization of other proteins [13]. The mass of histones is proportional to the mass of DNA in the sample, which in turn depends on the number of cells. 

### 2.3. Aspects of Proteome Variation

The abundance of proteins in the same population (tissue) can vary from cell to cell. Also, they can be present in different forms (proteoforms), interact with other molecules, or be in different locations. In clinical aspects, these variations have implications for cancer research and cancer therapy, in which a drug’s impact may vary due to variations in the proteome or the tumor heterogeneity [36]. One reason for proteome variability can be changes during the cell cycle progression. As much as 40% of cell proteins are cell-cycle dependent. Most of them exhibit changes in cellular localization, but ~11% change in protein level [37]. There is a high abundance variation for some proteins when entering the cell in M-phase, as shown by Beck et al. [38]. The proteins involved in cell cycle processes, nuclear division, mitosis, and microtubule cytoskeleton organization were increased in copy numbers up to 200 times, while multiple metabolic processes slowed down simultaneously [38]. Amazingly, the total copy number and balance between proteins with different copy numbers was not changed. For instance, see Figure 2, in which the distribution of spots is symmetrical around the diagonal [38]. Besides cell-cycle dependence, there is a general issue of cellular proteome heterogeneity. The proteome is composed of multiple proteoforms with variable abundance and diverse distributions inside different cells. This complexity is tightly regulated through the sophisticated molecular network called the protein homeostasis, or proteostasis. 

Proteostasis is a biological mechanism that controls biosynthesis, processing, folding, trafficking, and degradation of proteins in vivo. This mechanism contains over 1400 proteins with different functions. These proteins include chaperones, components of the degradation pathway, stress response enzymes, and numerous members of signaling pathways [39]. They are involved in proteome remodeling according to environmental conditions. Accordingly, the levels of all proteoforms that compose the cellular proteome should be properly presented.

For these reasons, single-cell MS analysis is the method of choice, especially when working with complex biological tissues [40]. It is constantly evolving, but because of sensitivity issues, the data volume generated by this method is still more limited than that produced by the widely used shotgun protocol [41]. Recent advances in sample processing, separation, and MS instrumentation now make it possible to quantify ~1000 proteins from individual mammalian cells [42]. Though the levels of these proteins vary between different cells, protein abundance distribution in single cells follows the distribution in the cell population [42,43]. Accordingly, the quantitative balance inside the cellular proteome is strictly regulated by the mechanisms of proteostasis.

### 2.4. Aspects of Sensitivity 

Bioinformatics approaches have revealed millions of proteoforms in a single human cell [2,5], but the numbers obtained experimentally so far are at least 100 times smaller. This gap will not be filled by new data very soon, considering the size of the human proteome and the sensitivity of the available proteomics methods. 

Many panoramic proteomics studies of different human cells or tissues have been performed, and several drafts of proteome maps of the human body have been published [44,45,46,47]. Though these maps are not complete, the massive volume of information obtained allows detailed and deep analysis. Here, protein (proteoform) abundance and the dependence of the number of proteins (proteoforms) on their abundance is an essential point of analysis. A special review of the sensitivity of proteomics methods and the evaluation of absolute copy numbers of proteins in a single cell was published by Orsburn [48]. In this paper, untargeted proteomics data were analyzed using the mean absolute copy number of proteins in a single cancer cell to measure instrument performance. Higher-sensitivity instruments generate more detected proteins and distributions with a lower maximum or median copy number. In the panel of the analyzed datasets, the log numbers of these go from 6.17 (data from 618 proteins) to 4.2 (data from 14,179 proteins) [48]. 

Based on multiple datasets, the technical aspect of instrument sensitivity is a factor in protein detection in areas of low abundance [48,49]. It was clearly shown, using the same sample but different chromatography times in ESI LC-MS/MS analysis, that the number of proteins in low abundance areas is often greatly underestimated [49]. Because of these limitations, the distributions of the number of proteins according to their copy numbers often are bell-shaped, and contain a peak corresponding to the proteins with the greatest copy numbers [48,49]. This situation can be observed more precisely by comparing more datasets and normalizing detected proteins not only to copy numbers but also to the %V, where V is the sum of all protein (proteoform) abundance (Figure 3). 

Such proteomics normalization is usually used in the analysis of images produced by 2DE and allows comparing cells with different volumes. Initially, all protein abundance is summed (V). Then each protein’s relative abundance (%V) is calculated, and proteins are grouped according to their descending values. Because of the broad range of abundance, a log scale was used. Additionally, a ranking of protein copy numbers was obtained using the data from Bekker-Jensen et al. for HeLa (Figure 3) [51]. In the case of normalization by copy numbers, it is important to remember that the copy number will be proportional to the cell volume [34], but this is revealed only when analyzing data from cells with small volumes, such as platelets [53]. 

The instrumental sensitivity issue can be observed more clearly when the data is presented in a slightly different way (Figure 4). Proteins are grouped according to their %V. The first group includes all proteins with relative abundance of 1% or greater, the second all proteins ≥ 0.5%, the third all proteins ≥ 0.25%, the fourth all proteins ≥ 0.125%, and so on. Using Excel, the numbers of proteins in each group (N) are plotted against relative abundance (%V). Visually, we can see that graphs of high abundance areas have very similar profiles. But at some point of abundance, depending on instrument sensitivity, the curves diverge. Therefore, the quantitative distribution of proteins in lower abundancy (sensitivity) areas must be carefully considered. Comprehensive analysis of available datasets can help us to reach final conclusions.

### 2.5. Aspects of Cancer

In response to various external or internal stimuli, cell proteomes undergo multiple changes. These changes can be analyzed by quantitative proteomics. In the case of cancer, quantitative protein analysis can be applied to cancer classification, diagnostics, drug selection, evaluation of drug resistance, assessment of therapeutic effects and toxicity, and the discovery of therapeutic targets and biomarkers. For example, differentiation of tumor subtypes is especially clinically important when choosing the type of treatment. Proteome-based classification may distinguish clinical features of lung cancers (adenocarcinoma or squamous cell cancer) and suggest therapeutic possibilities based on redox metabolism and immune cell infiltrates [54]. For example, the data show the importance of cadherin 2 in angiogenesis and highlight its potential both for antiangiogenic treatment and as a candidate prognostic marker for adenocarcinoma [55]. Furthermore, the Clinical Proteomic Tumor Analysis Consortium (CPTAC) aims to accelerate the understanding of the molecular basis of cancer through the application of proteomic technologies and workflows to clinical tumor samples [56]. Already, proteomic profiling based on quantitative mass spectrometry can categorize molecular subtypes for the propagation of 532 cancers [57]. SWATH/DIA-MS (State-of-the-art sequential window acquisition of all theoretical fragment ion/data-independent acquisition mass spectrometry) provides a promising supplement for stable classification of ovarian cancer subtypes [58].

A large amount of information was obtained in the tumor biomarker studies. Typically, paired tumor and adjacent tissue samples from patients and healthy individuals are prepared, digested into peptides, and analyzed using ESI LC-MS/MS. After quantification and filtration, many potential tumor biomarkers are selected based on their upregulated or downregulated level. The Human Protein Atlas (https://www.proteinatlas.org/, accessed on 25 April 2023) is a resource that contains much of that information. The Human Protein Atlas provides information on levels of gene and protein expression in the tissues and blood of patients with various diseases and highlights the proteins associated with these diseases. Importantly, in the case of cancer there is a panel of overexpressed proteins and a panel of downregulated proteins specific to each type of cancer. Therefore, during malignant transformation, the total amount of protein in the cell is not altered as much as the distribution profiles of protein/proteoform abundance. 

### 2.6. Analysis of Proteomics Datasets

There are few proteomics datasets in which absolute quantitation of the detected proteins in normal and cancer cells is available. Several publications were considered as sources of information about the abundance of proteins or proteoforms in different tissues or cell lines [26,27,44,45,46,47,50,52]. Some of them were used for the generation of Figure 3 and Figure 4 [44,45,46]. As shown in Figure 3 and Figure 4, a classical 2DE analysis of spot (proteoform) abundance is the least sensitive and allows a reliable quantitation of only about 1000 spots/proteoforms. The advantage of this quantitation (2DE spots) is that it is performed directly according to the estimation of protein mass by staining. The combination of 2DE with ESI LC-MS/MS dramatically improves the sensitivity: the number of detected proteins by sec2DE is increased to more than 5000 and the number of proteoforms to more than 20,000 (Table 2). 

The datasets from these publications were extracted and analyzed in the same way as in Figure 4. The linear scale was used to better apply a trend line. In all cases, the power function is the most appropriate trend for 2DE spot distribution (coefficient of determination R^2^ is from 0.91 to 0.98) (Table 2). Ideally, each spot should represent a single proteoform so that “spot abundance” is a synonym of “proteoform abundance”. The situation is more complicated in practice, and a single spot may accommodate many different proteoforms [26,27]. Usually, one proteoform is dominant and represents a significant proportion of the spot (at least 70%) [26,27]. An alternative and reliable way to evaluate proteoforms is to combine 2DE with ESI LC-MS/MS (a sectional 2DE with following ESI LC-MS/MS), which makes a more precise evaluation of proteoforms possible (Table 2) [28,52,60]. In all cases, the graphs (and formulas) are quite similar for all types of cells and tissues (Table 2).

The additional analysis of datasets from panoramic studies of multiple samples makes this observation even more reliable (Table 3, Figure 5). In a study by Wang et al. in which label-free mass spectrometry was used, 13,640 proteins from 29 healthy human tissues were quantified [45]. In a study by Jiang et al., 12,627 proteins across 32 normal human tissues were quantified using the TMT labeling method and TS scores for tissue-enrichment analysis [44]. A dataset produced by Kim et al. from the draft map of the human proteome contains the proteomic profiling of 30 histologically normal human samples, including 17 adult tissues, 7 fetal tissues and 6 purified primary hematopoietic cells, and resulted in identification of proteins encoded by 17,294 genes [46]. A mass-spectrometry-based draft of the human proteome fluids (human body map) published by Wilhelm et al. contains data from experiments involving 47 human tissues, cell lines, and body fluids [47]. A total of 18,097 proteins were identified in this study. Doll et al. have built a healthy human heart proteome by measuring 16 anatomical regions and three major cardiac cell types using high-resolution mass spectrometry-based proteomics. They quantified over 10,700 proteins in this tissue [61]. According to the detailed analysis, we can confirm that the dependence of the number of proteoforms on their abundance is described by the power function [59]. 

The available data should also be considered in terms of sensitivity. As previously mentioned, and clearly represented in Figure 4, in all MS datasets, the left shoulder or decline of the line is a result of lower instrumental sensitivity in the lower abundance area. Accordingly, data in this area cannot reliably represent abundance distribution. If we remove this area from our analysis (data marked with ¹), the final graphs are very similar to other charts (Table 3). Importantly, including these underestimated numbers of low-copy proteins/proteoforms in analysis and graph building can distort the final formulas presented in Table 3. For instance, according to the complete dataset presented by Bekker-Jensen et al. (14,000 proteins), the equation of abundance distribution is as follows: y = 41.443x^−0.553^(1)

But taking for analysis the first 6200 proteins from this dataset, which represent 98.5% of the sum protein mass, we get the equation: y = 12.004x^−0.931^(2)

Therefore, if the graphs are built based on the most reliable areas, they are very similar (Table 1 and Table 2). Accordingly, we can say that the final general formula is: y = Ax^−1^(3)
where y stands for the number of proteoforms (N) and x stands for the relative abundance of a proteoform that can be expressed as a percentage of the total mass (%V). According to the presented data, the coefficient A can be 5.5 to 15.4 (Table 3). This range may be a result of the sample preparation or of instrument peculiarities, as the data performed in different labs can produce different coefficients (Table 3). 

Finally, based on analysis of the available datasets, we can say that the proteoform abundance distribution in different cancer and normal human cells follows the same power function (3) or Zipfian probability distribution [59]. Zipf’s law has been identified in physics, biology, and the social sciences [62,63,64,65]. This law states, in particular, that the frequency of any word in a language is inversely proportional to its rank in the frequency table [62].

So far, researchers have only been able to hypothesize about the specificity and ubiquity of Zipfian distribution. Many theories have been proposed. In particular, it was suggested that the distribution can be the inevitable outcome of a very general class of stochastic systems [65,66]. An interesting explanation about the abundance of expressed genes and proteins was presented by Furusawa and Kaneko [63]. Using an abstract model of a cell with simple reaction dynamics, they showed that this power-law behavior in the chemical abundance generally appears when the reaction dynamics lead to a faithful and efficient self-reproduction of a cell. Therefore, these findings provide insights into the nature of the organization of complex reaction dynamics in living cells [63]. In the case of the human proteome, we can also say that distribution here reflects the functionality of different proteins/proteoforms and their abundance within the proteome. On the one hand, a human cell needs a high copy number (millions) of only a few proteins, such as actin or tubulin, for its structural organization. However, only a few copies each of many thousands of proteoforms are involved in processes such as signaling or protein turnover. 

There is also a practical aspect of Zipfian probability distribution of proteoforms inside the cell. We can apply the Formula (3) for proteoform number calculation to get an answer to the question “how many human proteoforms are there?” [9]. If the Formula (3) can be applied to the mass of proteoforms inside the cell, then the calculation task looks definite—insert the minimal value of the %V and calculate N. The only problem is that we don’t know to which proteoform population (%V or copy number) we should extrapolate the calculations. If it is one copy, or 10¯⁸ %V, then according to Formula (3), an average human cell has ~1 billion proteoforms.

Interestingly, if we look at the platelet, which has a volume ~400 times smaller than the average human cell (Table 1), the protein abundance of ~0.00005 %V corresponds to the range of single copies. In this case, according to Formula (3), N = 200,000. Therefore, there are ~200,000 different proteoforms in a single platelet. Hence, knowing the formula of proteoform abundance distribution inside the cell, we can get an impression about the whole cellular proteome organization and calculate the proteoform number.

## 3. Conclusions

Protein homeostasis (proteastasis) implies that the quantitative qualities of the human cellular proteomes persist in the stable state. Even though the cells have different volumes and different specialties, they have very similar quantitative characteristics to the proteomes. The proteoform abundance distribution in normal and cancerous human cells follows the same power function or Zipfian probability distribution [59]. In the case of cellular proteomes, this means that in a cell, the number of different proteoforms is inversely proportional to their abundance. 

## Figures and Tables

**Figure 1 ijms-24-08524-f001:**
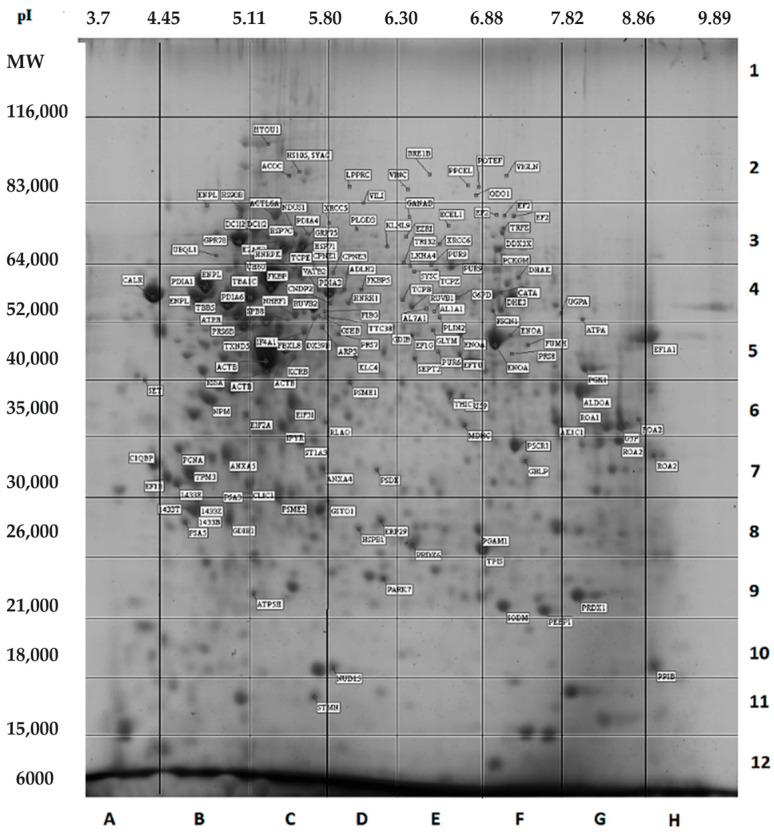
2DE map of HepG2 proteins. Spots with proteins identified by MALDI TOF-MS are annotated. Gel image analysis was performed using the program ImageMaster 2D Platinum 7.0 (GE Healthcare, Pittsburgh, PA, USA). Sections (marked with letters and numbers) in the 2D gel selected for following ESI LC-MS/MS analysis are shown. Reprinted with permission from [28].

**Figure 2 ijms-24-08524-f002:**
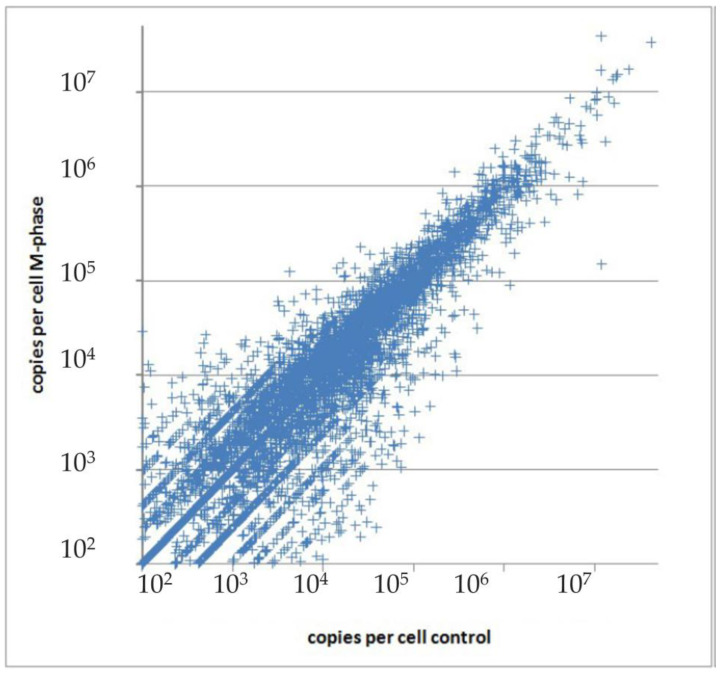
Logarithmic scale plot of protein copies per cell measured in non-synchronized and nocodazole-treated (M-phase) cells. Reprinted from [38].

**Figure 3 ijms-24-08524-f003:**
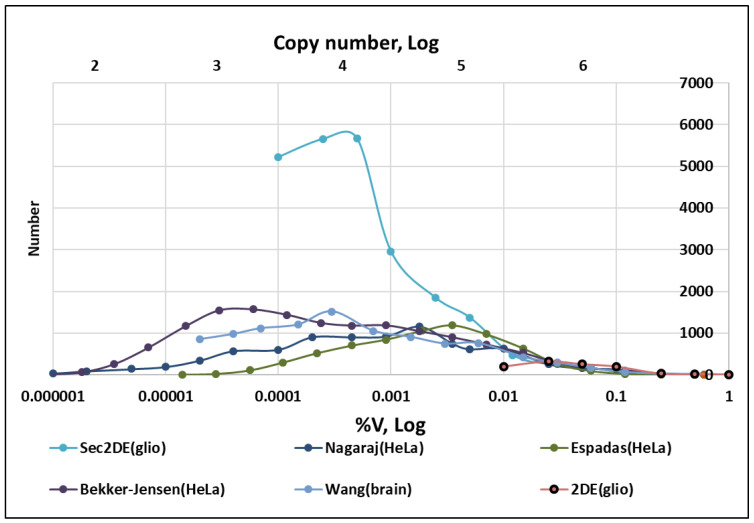
Examples of dependence of the number of detected proteins/proteoforms on their abundance. A distribution of abundance of proteins/protein groups/proteoforms is taken from datasets published by Espadas et al. [49], Wang et al. [45], Nagaraj et al. [50], Bekker-Jensen et al. [51], Naryzhny et al. (sec2DE(glio) and 2DE(glio)) [52]. The abundance of proteins/proteoforms was estimated using MS. The intensities of the stained spots were analyzed only for 2DE(glio).

**Figure 4 ijms-24-08524-f004:**
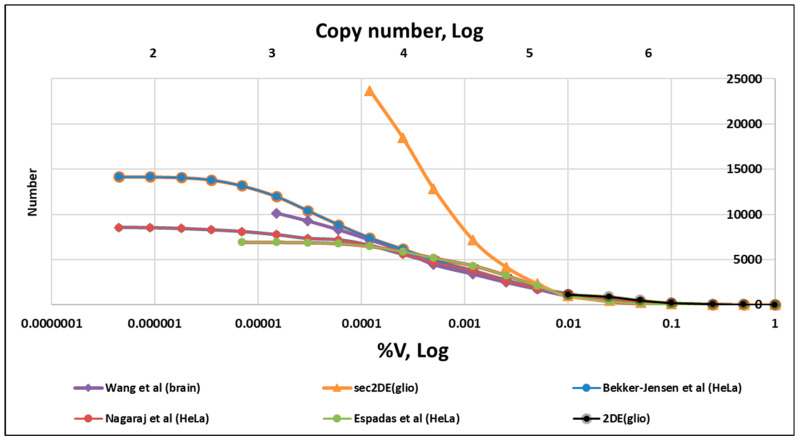
Examples of dependence of the number of all detected proteins/proteoforms on the abundance limit. The datasets of HeLa proteins produced in three different labs [49,50,51], brain extracts [45], and a classical 2DE analysis (spots) or sectional 2DE (sec2DE) analysis of proteoforms (glioblastoma cells) were used [52].

**Figure 5 ijms-24-08524-f005:**
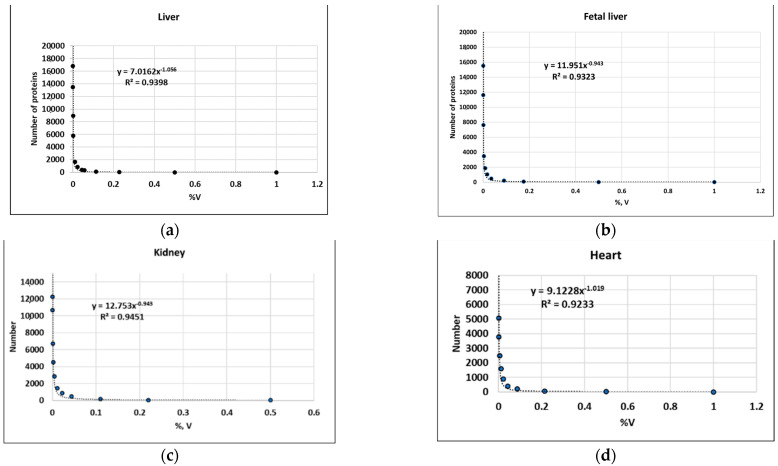
Examples of dependence of the number of proteins on their abundance (normalization in %V), when only extracts were analyzed by ESI LC-MS/MS. (**a**–**c**) Kim et al. [46], (**d**) Jiang et al. [44].

**Table 1 ijms-24-08524-t001:** Characteristic average volumes of human cells of different types.

Cell Type	Average Volume (μm^3^)	BNID [30] ¹
Platelet	10	[33]
Sperm cell	30	109,891
Red blood cell	100	107,600
Lymphocyte	130	111,439
Neutrophil	300	108,241
Beta cell	1000	109,227
Enterocyte	1400	111,216
Fibroblast	2000	108,244
HeLa, cervix	3000	103,725
Hepatocyte	3400	[32]
Osteoblast	4000	108,088
Cardiomyocyte	15,000	108,243
Fat cell	600,000	107,668
Oocyte	4,000,000	101,664

¹ ID numbers are from the database BioNumbers (BNID) or the reference number.

**Table 2 ijms-24-08524-t002:** Equations of dependence of the number of spots/proteins/proteoforms on their abundance in different cancer cells or a normal liver. The analysis is based on 2DE separation and 2DE maps.

Sample	Equation	Number	Reference
Glio (2DE spots)	y = 13.185x^−1.085^ R^2^ = 0.9261	1000	[51]
Glio (sec2DE)	y = 13.653x^−0.889^ R^2^ = 0.9845	24,000	[51]
HepG2 (2DE spots)	y = 17.459x^−1^ R^2^ = 0.9776	1300	[27]
HepG2 (sec2DE)	y = 9.99x^−0.984^ R^2^ = 0.9758	20,000	[27]
Liver (sec2DE)	y = 11.549x^−0.98^ R^2^ = 0.8952	15,000	[53]
Liver (2DE spots)	y = 13.452x^−1.16^ R^2^ = 0.9113	700	[53]
MCF7 (2DE spots)	y = 14.487x^−1.06^ R^2^ = 0.9792	700	[59]

**Table 3 ijms-24-08524-t003:** Equations of dependence of the number of proteins on their abundance in different human tissues or cells. Panoramic MS-analysis.

Sample	Equation	Number	Reference
Liver	y = 7.0162x^−1.056^ R^2^ = 0.9398	16,000	[45]
Fetal liver	y = 11.951x^−0.943^ R^2^ = 0.9484	16,000	[45]
Liver	y = 10.955x^−0.958^ R^2^ = 0.9007	5000	[43]
Liver	y = 7.2197x^−1.047^ R^2^ = 0.8961	5500	[43]
Adrenal	y = 7.2765x^−1.003^ R^2^ = 0.8627	7000	[43]
Adult Adrenal	y = 6.6905x^−1.051^ R^2^ = 0.9225	15,000	[45]
Adult Colon	y = 13.745x^−0.915^ R^2^ = 0.9766	15,000	[45]
Colon	y = 11.996x^−0.944^ R^2^ = 0.976	5000	[43]
Adult Esophagus	y = 15.288x^−0.876^ R^2^ = 0.9544	9000	[45]
Frontal Cortex	y = 12.544x^−0.953^ R^2^ = 0.9297	16,000	[45]
Adult gallbladder	y = 14.781x^−0.911^ R^2^ = 0.9727	10,000	[45]
Adult Pancreas	y = 12.281x^−0.948^ R^2^ = 0.9537	17,000	[45]
Pancreas	y = 12.537x^−0.894^ R^2^ = 0.8619	7000	[43]
Prostate	y = 12.246x^−0.939^ R^2^ = 0.973	4000 ¹ (11,000)	[44]
Adult Prostate	y = 14.466x^−0.916^ R^2^ = 0.9773	17,000	[45]
Adult Rectum	y = 11.495x^−0.932^ R^2^ = 0.9755	17,000	[45]
Adult Retina	y = 5.8187x^−1.079^ R^2^ = 0.9095	19,000	[45]
Spinal Cord	y = 11.821x^−0.946^ R^2^ = 0.9288	15,000	[45]
Adult Testis	y = 8.169x^−1.045^ R^2^ = 0.8192	20,000	[45]
Testis	y = 11.105x^−0.882^ R^2^ = 0.8996	9000	[44]
Fetal Testis	y = 5.439x^−1.092^ R^2^ = 0.9224	15,000	[45]
Placenta	y = 9.3737x^−0.998^ R^2^ = 0.9267	11,000	[45]
Kidney	y = 5.9506x^−1.075^ R^2^ = 0.9228	12,000	[45]
Heart	y = 15.719x^−0.927^ R^2^ = 0.9755	1500	[44]
Heart	y = 9.1228x^−1.019^ R^2^ = 0.9233	5000	[43]
Heart	y = 12.319x^−0.893^ R^2^ = 0.9824	13,000	[45]
Aorta	y = 12.254x^−1.009^ R^2^ = 0.9655	1200	[61]
Aortic valve	y = 17.85x^−0.698^ R^2^ = 0.8931	6800	[61]
Stomach	y = 10.254x^−1.017^ R^2^ = 0.8661	5000	[43]
Stomach	y = 15.361x^−0.905^ R^2^ = 0.9698	4000	[44]
Thyroid	y = 9.698x^−1.023^ R^2^ = 0.9185	5000	[43]
Muscle	y = 11.563x^−0.974^ R^2^ = 0.9422	3500	[43]
Muscle	y = 13.174x^−0.994^ R^2^ = 0.9409	9000	[44]
Brain	y = 10.672x^−0.985^ R^2^ = 0.8955	6000	[43]
Fetal brain	y = 8.584x^−0.981^ R^2^ = 0.9314	15,000	[45]
Lung	y = 9.2953x^−1.001^ R^2^ = 0.9583	12,500	[45]
Lung	y = 8.5254x^−1.023^ R^2^ = 0.6913	6000	[43]
Ovary	y = 7.3857x^−1.053^ R^2^ = 0.931	19,000	[45]
Fetal ovary	y = 7.4986x^−1.045^ R^2^ = 0.9368	17,000	[45]
Ovary	y = 9.8454x^−0.929^ R^2^ = 0.9009	6800	[43]
Platelets	y = 13.949x^−0.949^ R^2^ = 0.9909	3600	[52]
Platelets	y = 7.3257x^−1.127^ R^2^ = 0.9575	11,300	[45]
Uterus	y = 7.7271x^−1.059^ R^2^ = 0.9477	6000	[43]
B cells	y = 6.5677x^−1.051^ R^2^ = 0.9319	17,000	[45]
CD4 Cells	y = 7.5448x^−1.051^ R^2^ = 0.9533	14,000	[45]
NK Cells	y = 7.8551x^−1.029^ R^2^ = 0.9616	16,000	[45]
HeLa	y = 6.9393x^−0.963^ R^2^ = 0.9312	6000 ¹ (7000)	[48]
HeLa	y = 13.715x^−0.931^ R^2^ = 0.9453	4700 ¹ (10,200)	[49]
HeLa	y = 12.004x^−0.931^ R^2^ = 0.9187	6200 ¹ (14,000)	[50]

¹ Number of proteins taken for calculations.

## Data Availability

The data supporting the reported results can be found in the cited papers.

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
