# Peer review of "Quantitative Aspects of the Human Cell Proteome"

_ijms, 2023, doi:10.3390/ijms24108524_

Round 1
Reviewer 1 Report (New Reviewer)
The author explores the quantitative information originated from large-scale panoramic experiments where, in particular, 2-dimensional gel electrophoresis was used. The author finds that the distribution of the proteome was constant in all human tissues or cell lines analysed, and obeys the Zipf´s law.
major comments:
This review is interesting and well written. The only problem is that this review would be in the Special Issue entitled Proteomics and Its Applications in Cancers 2.0. However there is in this review from very little to nothing about the applications in cancers... There are merely few facts on cell lines scattered across the manuscript. I would suggest a large modification to the review so that the application of proteomics and Zipf´s law in cancers is debated, at least as a point.
Author Response
Answer – Thank you very much for the comments. According to your recommendation, a chapter “Aspects of cancer” was added.
Reviewer 2 Report (New Reviewer)
The manuscript titled “Quantitation aspects of the human cell proteome” reviews an important topic in proteomic analysis dealing with protein quantitation and their relative abundances in cells. The manuscript is clearly organized and analyses datasets from the literature to revise and present general features of the proteome based on correlation of protein copy numbers, relative abundance (%V) and number of protein identifications, stressing the difficulties in characterizing the low abundance elements. The manuscript outlines the importance of single cell proteomic analysis that offers more reliable results, if the instrumentation and/or proteomic methodology used is able to provide the suitable sensitivity. The elaboration of different cell types datasets and the description of proteostasis following the Zipf’s low is interesting, however as stated in the manuscript conclusion, stable conditions are required. Therefore changes in environmental conditions, different physiological states (i.e. age) and epigenetic alterations, diseases, are at the basis of clinical proteomics studies and/or cell proteomic profiling and are to be considered in addition to different instrument sensitivity of proteins detection and characterization. The different methods of protein extraction and preparation should be also considered and discussed in the datasets presented as an important parameter to be evaluated in data comparison. The application of the discussed equations in quantitative proteomic studies should be therefore more delineated and emphasized in addition to the strategy to include all variations that alter, especially, the less abundant proteins.
The manuscript should be checked for typing errors (i.e., line 35, this instead of his at line 191, ESI instead of ECI at line 246.
Other comments:
-line 87 the reference indicated is [24] but in the legend of Figure 1 is cited the reference [28]. Was that a mistake?
-line 205: what is “times” referred to in “different chromatographic times”? It is not clear.
-line 217: (Fig.2) should be replaced by Fig.3 instead?
: Fig.4 legend: similarly to Figure 3 legend in the graph, the reference should be added of the type of cells analyzed.
-lines 239-240: The references of the datasets available should be added.
Table 2: can higher R2 values mean better correlation also of the low abundant proteins data?
Author Response
Answer – Thank you very much for your comments. I am sorry, but seems it was a confusion about the stable conditions. You say…as stated in the manuscript conclusion, stable conditions are required. Sorry, may be it is my fault. The statement “The protein homeostasis (proteastasis) implies that quantitative qualities of the human cellular proteomes persist in the stable conditions” is about the stable conditions of the quantitative parameters of the proteomes not the stable conditions of the environment. I rephrased the statement in the conclusion “The protein homeostasis (proteastasis) implies that quantitative qualities of the human cellular proteomes persist in the stable state”.
Anyway, a chapter about cancer was added in the manuscript.
The manuscript should be checked for typing errors (i.e., line 35, this instead of his at line 191, ESI instead of ECI at line 246.
Answer - corrected
Other comments:
-line 87 the reference indicated is [24] but in the legend of Figure 1 is cited the reference [28]. Was that a mistake?
Answer – No, it’s not a mistake
-line 205: what is “times” referred to in “different chromatographic times”? It is not clear.
Answer - different chromatographic time in LC MC is a result of usage of columns with different length. Accordingly - longer column – longer time of chromatography – better analysis – more proteins are detected
-line 217: (Fig.2) should be replaced by Fig.3 instead?
Answer – Yes, it was a mistake. Corrected
: Fig.4 legend: similarly to Figure 3 legend in the graph, the reference should be added of the type of cells analyzed.
Answer – The references were added
-lines 239-240: The references of the datasets available should be added.
Answer – The references were added
Table 2: can higher R2 values mean better correlation also of the different chromatographic times data?
Answer – I don’t think so. I think the correlation depends on the quality of the datasets. If the analysis was performed by the same mass spectrometer but with different chromatographic times the quality is the same. Only number of detected proteins is different. As you can see, the R2 value does not correlate with the number of detected proteins.
Round 2
Reviewer 1 Report (New Reviewer)
The author debate the quantitative aspects of human proteome, including the quantitative infomation obtained from large-scale panoramic experiments. The author identifies Zipf's law in the distribution of proteome components.
Upon addition of a point concerning the aspects of cancer, the review has significantly improved and I have no furhter concerns.
Author Response
Thank you very much for your time and comments
This manuscript is a resubmission of an earlier submission. The following is a list of the peer review reports and author responses from that submission.
Round 1
Reviewer 1 Report
Naryzhny provides a nice review of the current understanding of the human proteome, and this review contains many nice details (e.g., size and extent of the proteome, variance among tissue types, etc.). The author also nicely highlights some key problems, for example, that bioinformatics based approaches indicate a single human cell contains millions of proteoforms while experimental techniques lack the sensitivity to verify such predictions. Unfortunately, the significance of the main push of the article, that proteome distributions follow Zipf's law, while demonstrated (Figs 4-6, Tables 2-3), was poorly described, especially with regard to why anyone should care if the distributions follow Zipf's law. This is a major concern.
Some other concerns:
1. It's not clear what "bottom-up" and "top-down" refer to; please provide a definition. And, why do "top-down" techniques have mass limitations that "bottom-up" techniques do not? Some background details would be helpful to the uninitiated. line 73
2. The Bradford protein assay uses a different form of coomassie blue than gel staining, correct? G-250 vs R-250. As such, it's probably not best to use the Bradford assay as evidence that gel staining provides a linear dose response. Moreover, the Bradford assay is a solution reading that is linear only in the dilute limit, again complicating an extrapolation to gel staining. line 82
3. There are some issue with sentence structure and grammar usage. I recommend professional editing.
Example: "This review analyzed the quantitative information obtained from several large-scale panoramic experiments, where high-resolution mass spectrometry-based proteomics in combination with liquid chromatography or two-dimensional gel electrophoresis." should be "This review analyzed the quantitative information obtained from several large-scale panoramic experiments, where high-resolution mass spectrometry-based proteomics in combination with liquid chromatography or two-dimensional gel electrophoresis were used to evaluate the cellular proteome." line 13
Example: "only relative quantification can perform." Can perform what? line 76
Example: "It should be pointed an important aspect here." ?? line 91
Example: "The six cell types comprise 97% ..." should be "Six of the cell types comprise 97% ..." line 112
Example: "Up today, many panoramic proteomics ..." should be "Up through today, many panoramic proteomics ..." line 168
Author Response
Answer to the reviewer.
First of all, I would like to thank you very much for the valuable comments and suggestions. Below are my answers, point by point.
Naryzhny provides a nice review of the current understanding of the human proteome, and this review contains many nice details (e.g., size and extent of the proteome, variance among tissue types, etc.). The author also nicely highlights some key problems, for example, that bioinformatics based approaches indicate a single human cell contains millions of proteoforms while experimental techniques lack the sensitivity to verify such predictions. Unfortunately, the significance of the main push of the article, that proteome distributions follow Zipf's law, while demonstrated (Figs 4-6, Tables 2-3), was poorly described, especially with regard to why anyone should care if the distributions follow Zipf's law. This is a major concern.
Answer. Thank you very much for your comments. I am sorry, but I cannot agree that proteome distributions according to Zipf's law is poorly described. I am not sure what else should be presented in the description in addition to the multiple examples (minimum 56 samples were analyzed!). I think that two important main things are clearly shown in the text – 1) in all human cells, the distribution of proteoforms is following the same rules, 2) this distribution is following the Zipf's law. I think that this is important, and people should care about it, as at least it allows to know how the proteome is organized.
Some other concerns:
- It's not clear what "bottom-up" and "top-down" refer to; please provide a definition. And, why do "top-down" techniques have mass limitations that "bottom-up" techniques do not? Some background details would be helpful to the uninitiated. line 73
Answer. Extra explanation was added in the text.
- The Bradford protein assay uses a different form of coomassie blue than gel staining, correct? G-250 vs R-250. As such, it's probably not best to use the Bradford assay as evidence that gel staining provides a linear dose response. Moreover, the Bradford assay is a solution reading that is linear only in the dilute limit, again complicating an extrapolation to gel staining. line 82
Answer. 2DE gels that was analyzed in the present manuscript were stained mainly by Coomassie R. But Coomassie G was used as well [54]. Actually, in terms of linear dose response, there is practically no difference between these forms of Coomassie. The wide linear range of protein staining in the gel by this dye was shown before [24,25]. The Bradford assay was mentioned here not as an evidence but just as another example of such a convinient qualities of Coomassie. Sorry, if it was not clear from the text.
- There are some issue with sentence structure and grammar usage. I recommend professional editing.
Example: "This review analyzed the quantitative information obtained from several large-scale panoramic experiments, where high-resolution mass spectrometry-based proteomics in combination with liquid chromatography or two-dimensional gel electrophoresis." should be "This review analyzed the quantitative information obtained from several large-scale panoramic experiments, where high-resolution mass spectrometry-based proteomics in combination with liquid chromatography or two-dimensional gel electrophoresis were used to evaluate the cellular proteome." line 13
Answer. Thank you very much for the correction. The text was edited.
Example: "only relative quantification can perform." Can perform what? line 76
Answer. The sentence was corrected.
Example: "It should be pointed an important aspect here." ?? line 91
Answer. The sentence was corrected.
Example: "The six cell types comprise 97% ..." should be "Six of the cell types comprise 97% ..." line 112
Answer. The sentence was corrected.
Example: "Up today, many panoramic proteomics ..." should be "Up through today, many panoramic proteomics ..." line 168 .
Answer. The sentence was corrected.
Reviewer 2 Report
Lines 14-15. Some kind of verb was omitted?
Line 82. Misspelled "dye".
Lines 112-114. Percentage numbers do not correlate with the total amount of 100%. Additional clarification is needed.
Lines 181 and 205. The number of traced lines on the chart and on the legend does not match. It is necessary to clarify this point or replace the colors with more different ones. Could not find/distinguish Wang and 2DE in the Fig. 2 and trace the 2DE in the Fig. 3.
Line 211. A different representation of the data in Figures 2 and 3 is claimed, although they differ only in the dimension of the ordinate at first glance. Too complicated descriptions are presented. More clear explanations are needed about the differences of the two figures. Are they presenting spot and LC data or not? It’s not clear.
Line 233. It is not clear from the Table 2 where are the results of spot analysis and the results of LC analysis, except the numbers, which are presented in different ways for Glio/Hep and for Liver.
Line 281. There is no mention of Figure 6 in the text.
Author Response
Answer to the reviewer
First of all, I would like to thank you very much for the valuable comments and suggestions. Below are my answers, point by point.
Lines 14-15. Some kind of verb was omitted?
Answer. It was corrected
Line 82. Misspelled "dye".
Answer. It was corrected
Lines 112-114. Percentage numbers do not correlate with the total amount of 100%. Additional clarification is needed.
Answer. It was corrected. Thank you very much for this observation.
Lines 181 and 205. The number of traced lines on the chart and on the legend does not match. It is necessary to clarify this point or replace the colors with more different ones. Could not find/distinguish Wang and 2DE in the Fig. 2 and trace the 2DE in the Fig. 3.
Answer. It was corrected. Thank you very much for this observation.
Line 211. A different representation of the data in Figures 2 and 3 is claimed, although they differ only in the dimension of the ordinate at first glance. Too complicated descriptions are presented. More clear explanations are needed about the differences of the two figures. Are they presenting spot and LC data or not? It’s not clear.
Answer. Extra explanation was added. In Fig.2 each point represents the number of proteins/proteoforms that are having the similar abundances in this area of %V. For instance, for sec2DE(glio), the point with coordinates (3000/0.001) means that there are 3000 proteoforms having %V in the range 0.001-0.002 only. But in Fig 3, each point represents all proteins/proteoforms having the particular %V or smaller. For instance, for sec2DE(glio), the point with coordinates (7150/0.001) means that there are 7150 proteoforms having %V 0.001 or less.
Line 233. It is not clear from the Table 2 where are the results of spot analysis and the results of LC analysis, except the numbers, which are presented in different ways for Glio/Hep and for Liver.
Answer. More information was added in the Table 2.
Line 281. There is no mention of Figure 6 in the text.
Answer. It was corrected